# The Effect of Jackfruit Skin Powder and Fiber Bleaching Treatment in PLA Composites with Incorporation of Thymol

**DOI:** 10.3390/polym12112622

**Published:** 2020-11-07

**Authors:** Muhammad Najib Ahmad Marzuki, Intan Syafinaz Mohamed Amin Tawakkal, Mohd Salahuddin Mohd Basri, Siti Hajar Othman, Siti Hasnah Kamarudin, Ching Hao Lee, Abdan Khalina

**Affiliations:** 1Department of Process and Food Engineering, Faculty of Engineering, Universiti Putra Malaysia, Serdang 43400, Selangor, Malaysia; najibmarzuki@yahoo.com (M.N.A.M.); salahuddin@upm.edu.my (M.S.M.B.); s.hajar@upm.edu.my (S.H.O.); 2Laboratory of Halal Services, Halal Products Research Institute, Putra Infoport, Universiti Putra Malaysia, Serdang 43400, Selangor, Malaysia; 3School of Industrial Technology, Faculty of Applied Sciences, Universiti Teknologi Malaysia (Mara), Uitm Shah Alam, Shah Alam 40450, Selangor, Malaysia; sitihasnahkam@uitm.edu.my; 4Institute of Tropical Forestry and Tropical Products, Universiti Putra Malaysia, Serdang 43400, Selangor, Malaysia; khalina@upm.edu.my

**Keywords:** jackfruit skin powder fibre, characterization, antimicrobial activity, bleaching treatment, polylactic acid

## Abstract

Food packaging has seen a growth in the use of materials derived from renewable resources such as poly(lactic acid) (PLA). However, the initial costs to produce bioplastics are typically high. Tropical fruit waste as naturally sourced fibres, such as jackfruit skin, can be used as a cost-reducing filler for PLA. The main objective in this study is to fabricate a low-cost natural fibre-reinforced polymer that potentially applies in packaging with the aid of bleaching treatment. The treatment shows a rougher surface fibre in Scanning electron microscopy (SEM) micrographs and it is expected to have better mechanical locking with the matrix, and this is found similar with a Fourier-transform infrared spectroscopy (FTIR) analysis. Unfortunately, fibre insertion does find low tensile performances, yet bleached-fibre composites improved its performance significantly. A similar situation was found in the thermal characterization where a low-thermal stability natural fibre composite has lower thermal behaviour and this increased with bleaching treatment. Besides, bleached-fibre composites have a longer service period. Besides, a 15 wt% thymol insertion inhibits the growth of Gram-positive bacteria in the composites and the non-treated fibre composite has better thymol effects. The 30 wt% of the bleached-fibre insertion composite has a high potential to reduce the cost of bioplastic products with minimum alterations of overall performances.

## 1. Introduction

Natural fibre-reinforced polymer composites are not something new nowadays. Natural fibre reinforcement has been proven to be comparable with synthetic fibre composites and is commonly applied in major advanced industries, such as aerospace, automotive, medical and packaging [1,2,3]. Other than technical performances, natural fibre reinforcement also promotes the use of green materials which are harmless to the environment, almost zero in cost, wheel the social economics of low-income families as well as prolong fabrication tools’ life and reduce workers’ health issues [4]. Together with all factors, natural fibre usage has been intensively increasing to replace synthetic fibres. Natural fibres are a big family group and consist of hundreds of types of natural fibres. However, only a few major types of the natural fibre undergo enormous research developments. The other types of natural fibres could not attract the attention of researchers or industries due to many factors, such as the limitation of geology conditions, growth and/or fibre extraction difficulties [5]. These fibres failed to create a complete supply–demand cycle and thereby further knock back by the public.

Jackfruits or *Artocaprus heterophyllus* are a common tropical fruit that is usually consumed in the Southeast Asia region. It originated in the rain forests of the Western Ghats in the southwestern part of India and is now widely distributed in tropical countries such as Brazil, Thailand, Indonesia, India, the Philippines and Malaysia [6]. Jackfruits are composed of several berries of yellow pulp and brown seeds encased in a hard shell and are rich in carbohydrates, complex B vitamins and minerals. However, only 15–20% of the fruit is used as food, which can be cooked, baked or roasted on coals. The waste produced is 65–80% of the total weight of the jackfruit [7]. This waste can be potentially turned to benefit biomass through incorporation as a filler or reinforcement into a polymer matrix. Natural fibre reinforces polymer composites by several modes (long fibre, short fibre, textile, non-woven and powder forms) [8]. Jackfruit fibre reinforcement and/or fillers have been claimed potentially of use in polymer film packaging [9]. However, limited previous studies conducted on this fibre and hence make it one of the most under-utilised natural fibres

Jackfruit peel is commonly treated as waste; it consists of ca. 50% of α-cellulose which makes it suitable for microcrystalline cellulose production as well as low-cost filler for the production of biocomposite materials [10]. The amount of fruit wastes produced increases when the production of processed fruit products increases. Jackfruit wastes such as skin, straw and seed have no economic value, which can lead to environmental problems. However, this waste can potentially benefit biomass through incorporation as a filler into the polymer matrix. At present, the available information on the utilization of jackfruit skin as a filler in the composite system is still limited. The presence of this low-cost filler in polymer material may be able to improve the performance of virgin materials as well as reduce the density and lower the cost of end products [11,12].

Bleaching treatment is a well-known fibre treatment to improve fibre/matrix interfacial bonding. It modifies and activates the fibre structure using a hydroxyl group chemical to have a higher number of bonding sites for better adhesion. Bleaching treatments remove non-cellulosic components on the fibre surface and, most importantly, they are environmentally safe [13]. A previous study showed an improvement of mechanical properties when the fibres were subjected to bleaching treatment [14]. Enhanced interfacial adhesion is the reason behind this improvement. Besides, bleaching treatment also gives a whitening effect to the fibre and hence the appearance of the final product [15]. The whitening effect is a critical criterion for some applications where physical appearance is important, such as food packaging including film and paper.

Maintaining the food freshness is a difficult task. Antimicrobial properties inherent in food packaging could help to increase food shelf-life and thereby reduce food waste. The fresh food will be attacked by microbes and bring illness to consumers. The growth of microbes on foods’ surfaces shall be accelerated once they get water and oxygen. Unfortunately, it is very hard to eliminate all water and oxygen for fresh food. There are many natural antimicrobial agents that are suitable for food packaging available in the market, such as carvacrol, thymol and cinnamaldehyde, which inhibit noticeable antimicrobial activity [16]. However, thymol is one of the famous antimicrobial agents and it exhibits the highest in vitro antimicrobial activities against *E. coli* and *S. aureus* [17]. It is less water-soluble at neutral pH, but it dissolves well in organic solvents and alcohols. Petchwattana and Naknaen (2015) found thymol insertion in poly(butylene succinate) film has an effective microbes inhibition properties [18]. Marchese et al. (2016) reviewed the antibacterial and antifungal activities of the thymol, as a safe food preservative agent [19]. However, incorporation of the thymol fillers in composites found a deterioration of the composite’s mechanical performances yet a slight improvement in its thermal stability [20].

In previous studies, there are limited investigations on the use of jackfruits fibres for composite reinforcement and no past research conducted on the application of thymol as an antimicrobial agent in jackfruit fibre-incorporated poly(lactic acid) (PLA) composites [20,21,22]. In this study, a bleaching treatment shall be conducted on jackfruit fibre before mixing with PLA composite to show the effect of fibre treatment and reinforcement into PLA composite. The insertion of thymol into PLA and PLA composites also carried out in order to inhibit the growth of bacteria in vitro.

## 2. Materials and Methods

### 2.1. Materials

Ripe jackfruit from *Nagka Madu* (J33) species was purchased from a wholesale market (Selangor, Malaysia). The preparation of jackfruit skin powder (JSP) involved washing, chopping, drying, grinding and sieving. Polylactic acid IngeoTM, 7001D was provided in resin form by NatureWorks LLC (Minnetonka, MN, USA). The density, melting temperature range and glass transition temperature of PLA were 1.24 g/cm^3^, 145–160 °C and 59 °C, respectively.

The chemicals used for the bleaching or delignification process were sodium hydroxide (NaOH), acetic acid (CH_3_COOH) and sodium chlorite (NaClO_2_) with 80% purity, supplied from R&M Chemicals (Selangor, Malaysia). A natural antimicrobial agent, thymol is also known as a food additive and was purchased from R&M Chemical, Malaysia. Nutrient agar, nutrient broth and peptone water were purchased from Oxoid, Malaysia. Gram-positive bacteria, *Staphylococcus aureus* (*S. aureus*), were provided by the Institute of Bioscience, Universiti Putra Malaysia, Malaysia. The overall flow diagram is shown in Figure 1.

### 2.2. Preparation of Jackfruit Skin Powder (JSP)

The jackfruit was chopped in half in order to separate its flesh and skin. The jackfruit skin was then washed and oven-dried at 60 °C for 72 h. Next, the dried jackfruit skin was crushed using a grinder (Retsch SM200, Haan, Germany) and passed through a 0.5-mm-sized mesh. The sample was inserted into a vibratory sieve shaker (Retsch AS200, Haan, Germany) with a 250–500 µm mesh size to produce a constant size of powder. All the experimental works conducted at Universiti Putra Malaysia. 

### 2.3. Bleaching of Jackfruit Skin Powder (BJSP)

The bleaching treatment of JSP was carried out following the conditions in accordance with ASTMD1104–56 in order to produce holocellulose, which is primarily designed to remove lignin [21]. Initially, 10 g of the JSP was rinsed with distilled water to remove dust and foreign materials. This was followed by soaking in a 500 mL beaker containing 300 mL of hot distilled water. Then, the beaker was transferred into the water bath which was set at 70 °C. Next, 2 mL of acetic acid and 4 g of sodium chlorite were consecutively added every 1 h into the beaker for a total process time of 5 h. The mixture was stirred with a glass rod and covered with aluminium foil. The bleaching process was indicated by the colour change of the JSP from light brown to white. It was then washed and rinsed with distilled water until the yellow colour and the odour was removed.

### 2.4. Preparation of PLA/JSP and PLA/BJSP Composites with Thymol

The PLA and PLA composites were compounded by using a Brabender Plasti-Corder (Retsch AS200, Haan, Germany) internal mixer, which was equipped with a twin roller. The roller speed of the internal mixer was set at 50 rpm. The mixing process was carried out at 170 °C to ensure complete melting of PLA resins. The time taken to blend the composite in the mixer was about 8 min. The JSP was added into the hopper 2 min after the PLA has been melted in the chamber. The PLA resin was dried at 60 °C for 24 h prior to the compounding process. A hot press machine was used to fabricate the PLA and PLA composite sheets. The specimen was preheated at 160 °C for 4 min and pressed at the same temperature before cooling. A hand-held digital vernier calliper (B.C Ames Co., Framingham, MA, USA) was used for measuring composite sheets with a thickness of 1 to 2 mm. The composite formulation is shown in Table 1.

### 2.5. Composites Characteristics

#### 2.5.1. Infrared Spectroscopy

A Fourier-transform infrared spectroscopy (FTIR) Perkin Elmer Spectrum One FT-IR Spectrometer (Waltham, MA, USA), with the attenuated total reflectance (ATR) technique, was used to detect possible changes in the functional group of untreated and bleached JSP. All the spectra were recorded in the transmittance mode with a resolution of 4 cm^−1^ in the range of 400 to 4000 nm. Ten scans were performed for each acquisition.

#### 2.5.2. Scanning Electron Microscopy

Scanning electron microscopy (SEM) was performed to observe the surface topography of JSP, BJSP and 30-BJSP-10THY using SEM LEO 1455 VP, (Carl Zeiss AG, Oberkochen, Germany).

#### 2.5.3. Tensile Testing

The tensile test was performed using a 5-kN Universal Instron 3365 Testing Machine according to the standard ASTM D638 with a cross-head speed of 5 mm/min. Prior to the tensile test, the width and thickness of the specimens were measured by using a micrometre. Five specimens of the sample were cut using a dumbbell-shape cutter for each sample were tested and analysed. Tensile strength, elongation at break and tensile modulus was obtained from the plotted graph.

#### 2.5.4. Thermogravimetric Analysis

Thermal properties of sample were analysed using a thermogravimetric analysis. In this analysis, a Pyris 1 Thermogravimetric Analyser PerkinElmer was used. Firstly, the pan of the instrument must be cleaned so that no contamination occurred. All samples were heated from 50 to 600 °C with a heating rate of 10 °C/min (heating scan). Then, the degradation temperature of composites and evaporation temperature of thymol were obtained from the graph.

#### 2.5.5. Differential Scanning Calorimetry

The thermal properties of the composites also were measured by using differential scanning calorimetry (DSC) under an inert gas (nitrogen) atmosphere. The samples of composites were weighed and sealed in aluminium crucibles. The heating was performed over the range 30 to 300 °C at a rate of 10 °C/min and with a nitrogen flow rate of 20 mL/min.

#### 2.5.6. Composite Degradation Test

A qualitative study of the decomposition under composting conditions was performed on composite samples cut into pieces (20 mm × 14 mm × 1.3 mm). Samples were buried in a commercial compost at a 5-cm depth in perforated boxes and incubated at 58 °C. Aerobic conditions were maintained by mixing the compost periodically and by the addition of water to maintain a moisture content equivalent to 60% relative humidity. Samples were removed from the compost after 7 and 15 days, were immediately washed with distilled water to remove traces of compost, and then photographed.

#### 2.5.7. Antimicrobial Activity

In order to determine the effectiveness of thymol as an antimicrobial agent, a disc of diffusion test was conducted. A single strain of bacteria S. *aureus* was grown in nutrient broth and incubated at 37 °C for 24 h until the total bacterial count was 10^6^–10^7^ cfu/mL. A ten-fold dilution was then conducted where 9 mL peptone water was mixed with 1 mL of the test sample three times. In total, 100 µL of the test sample was then transferred via micropipette and spread onto a agar plate which was then incubated for 24 h with the PLA and the composite containing thymol, as well as tetracycline 30 antibiotic (as negative control) discs, was placed on the agar. The presence of a zone of inhibition was observed.

#### 2.5.8. Statistical Analysis

The data were analysed using Minitab 14.12.0 (Universiti Putra Malaysia, Malaysia) by one-way analysis of variance (ANOVA), followed by Tukey’s multiple comparison tests at a 95% confidence level.

## 3. Results and Discussions

### 3.1. SEM

The SEM micrograph in Figure 2 is to show and compare the fibre surface difference between (a) unbleached JSP and (b) bleached jackfruit skin fibre (BJSP). In the figure, it clearly shows a rougher fibre surface after bleaching treatment, showing an effective removal of non-cellulosic components. This provides a high expectation on better characterizations when applying BJSP in PLA composites, with good mechanical fibre locking with the matrix.

On the other hand, the incorporation of thymol in the BJSP/PLA composite found fibre pull-out during the fracture condition, as shown in Figure 3. The thymol, which was originally oil-based, acts as a lubricant between the fibre and matrix interface, weakening interfacial bonding and accelerating fibre pull-out [23]. Hence, a lower strength performance is suggested in this study.

### 3.2. FTIR

In order to confirm the effect of bleaching treatment on the jackfruit skin powder (JSP)’s, surface FTIR spectra of JSP and BJSP were obtained and are shown in Figure 4. In both JSP and BJSP spectra, there is a broad peak recorded at 3330 cm^−1^, indicating the presence of O–H groups as agreed by Ilyas (2017) [24]. This is contributed by cellulose, hemicellulose and lignin components, which are found rich in all types of natural fibres. However, there is a slight peak shift between the JSP and BJSP spectra. This suggested that after the bleaching process there is a slight change in O-H groups in the JSP.

On the other hand, an absorbance peak at 2890 cm^−1^ in the JSP spectrum is observed which corresponds to the C–H group stretching. This result could be in agreement with the findings of Rayung (2014) [14]. This peak was bluntly presented after the bleaching treatment, which specified the changes in C-H group components during chemical extraction. This also was supported by Jonoobi (2010) [25]. Moreover, a lower intensify peak was found at 1600 cm^−1^ for BJSP showing a successful bleaching treatment that removed the aromatic groups presented in lignin components [26].

### 3.3. Mechanical Properties

In this study, the pure PLA possessed a higher tensile strength (53.2 MPa) than all PLA composites with BJSP or JSP insertion, as shown in Figure 5a. A decrease in tensile strength resulting from an increase in fibre loading could be attributed to a weaker interfacial interaction between PLA and fibre, which resulted in an inefficient stress transfer [14]. Fortunately, the tensile strength of BJSP-filled PLA composites recorded significant increments with *p* < 0.05. This trend can be ascribed to the better interfacial adhesion between the fibre and matrix, due to removal of non-cellulosic components via bleaching treatments, and it is synchronised with the FTIR analysis and evidenced by the SEM micrograph [27]. On the other hand, a higher crystallinity index was found due to higher fibre contents and/or fibre treatments as reported in a previous study [28]. This higher crystallinity index was responsible for the higher strength performances. An insertion of 30 wt% jackfruit fibre is the optimum reinforcement loading for PLA polymer composites, where the highest tensile strength was recorded among all fibre reinforcement ratios, for both treated and untreated fibres. Excess fibre loading resulted in incomplete wetting due to insufficient matrices and hence lower strength capacity. Besides, there is a high possibility that excess fibre powders’ insertion leads to fibre agglomeration, creating non-homogeneous composites.

PLA is a stiff but brittle material and, with the addition of fibre, shows better tensile modulus values—fibre-reinforced composites present a stiffer material. All JSP or BJSP composites were found to have a higher tensile modulus than pure PLA and this is agreed by previous investigations (Figure 5b) [29]. The presence of fillers restricted the polymer’s chain mobility and thereby yielded a higher composite stiffness.

On the other hand, the elongation at break of the pure PLA shows the highest value (9%) compared to its composites (Figure 5c). Furthermore, the elongation at break decreased with increased fibre loading, regardless of treatment, but no significant changes from 10 to 30 wt% loading of powder was observed. This is because of the inherent rigidity of the JSP particles, leading to a loss of ductility in the PLA composites. At the same time, a lower elongation at break for BJSP composites was due to the enhanced interfacial adhesion, which led to the lower polymer chain mobility, another signal of successful treatment.

Figure 6a shows the tensile strength for PLA and 30BJSP composites with different thymol loading (5–15 wt%), to investigate the effect of thymol insertion. The strength of pure PLA is higher than the fibre-inserted composites, regardless of thymol loading. Observing the chart shows that as the thymol increases, the strength performances of the composites are lower. This observation was predicted and corresponds with a previous study [30]. For a lack of better words, the addition of thymol compounds found in essential oils resulted in a lowered interaction between PLA molecules and obstructs the polymer chain-to-chain interactions which ultimately causes a decrease in the tensile strength [31]. This situation was at its worst when natural fibres were inserted together in the composite system.

Figure 6b shows the tensile modulus for PLA polymer and 30BJSP composites with different thymol loadings (5–15 wt%). Insignificant changes of the tensile modulus were observed for all specimens except 15 wt% of thymol contents composite with *p* > 0.05. It also shows a very drastic decrease (82%) in its tensile modulus and suggests a shift from brittle to elastic properties. Thymol has been reported to exhibit a plasticizer effect when it is mixed into a polymer matrix. It is known to lower the intermolecular force and cause an increase in the mobility of polymeric chains, thus increasing the polymer’s flexibility [23,32]. This thus supports the results in this study where the insertion of thymol increases the flexibility of composites by reducing the tensile modulus.

Thymol reinforcement was also found to affect the elongation at break for specimens. Figure 6c shows the elongation at break for the PLA polymer and 30BJSP composites with different thymol loadings (5–15 wt%). By adding thymol, the elongation at break gradually increases significantly (*p* < 0.05), which suggests a higher elasticity of specimens, for both with and without fibre reinforcements.

The addition of fibre shows a decrease in the elongation at break compared to its PLA/thymol counterpart. This suggests that fibre reduces the elasticity of the composite, which is related to the previously mentioned statement that fibre is a stiff material that causes the composite to be stiffer and less flexible. As the thymol content reaches 15 wt%, the elongation at break increases drastically which may refer to thymol being a dispersing agent for the fibre which in return causes it to lose stiffness and thus increases in flexibility [33].

### 3.4. Differential Scanning Calorimetry (DSC)

Table 2 summaries the key data obtained from the analysis of the DSC thermograms of PLA and its composites. The onset of *T*_g_, *T*_cc_, *T*_m_ of the PLA composites containing JSP or BJSP showed no significant differences/slightly lower as compared to PLA polymer. The PLA polymer demonstrates double melting peaks with one higher dominant peak at higher temperatures (*T*_m2_). These melting peaks shifted slightly to a lower temperature for 10JSP yet recorded no difference for the 30BJSP specimen. The reason behind this is because the poor interfacial bonding of the fibre/matrix absorbs less heat energy before melting and this is found to be synchronised with the strength profile and lower mechanical properties for 10JSP. After this, the addition of fibre reinforcement increases heat absorption in order to melt. On the other hand, treated fibre-reinforced composites requiring a higher melting temperature are responsible for the better fibre/matrix interlocking mechanism that needs more energy to break down.

The insertion of natural fibres reduced the glass transition temperature, *T*_g_, and higher decrement for higher fibre contents. This was due to insufficient polymer wetting on the fibre surface, making its polymer chains slippery at lower temperatures. When comparing treated and untreated fibre reinforcements, the results were expected. Bleaching treatments remove non-cellulosic components on fibre surfaces, and later aids in increased physical entanglement with the PLA matrix, thereby resulting in higher *T*_g_ values [34]. On the other hand, fibres and thymol act as nucleating agents by showing evidence of earlier crystallization temperatures (*T*_cc_) for all PLA composites. This is found to be aligned with a previous PLA composite study [35].

Figure 7 represents the DSC thermograms of the PLA and its thymol composites. By increasing the thymol contents, it can be observed the a significant reduction in the *T*_g_ value can be associated with the plasticizing effect from thymol [23]. The destruction of polymer bonding by thymol made the composite more and more flexible—evidence from the elongation analysis. This disturbance was found to be even worse when introducing natural fibres into the thymol composite. Therefore, a lower glass transition and melting temperature were observed with the increase in thymol loadings.

### 3.5. Thermogravimetric Analysis (TGA)

The objective of a thermogravimetric analysis is to investigate the decomposition and degradation of composites at higher temperatures. In general, the first process of weight loss of fibre is attributed to the thermal degradation of lignin and hemicellulose. The next weight loss is associated to the decomposition of the α-cellulose present in the fibre. The thermogravimetric (TG) profiles of neat PLA and PLA composites containing JSP and BJSP, with loadings of 10 wt% and 30 wt% JSP, are shown in Figure 8a in the form of the weight loss as a function of temperature. The corresponding derivative weight-loss curve is also shown in Figure 8b to give a more detailed analysis of the TGA data to be made which provides the rate at which the different composites decompose.

The PLA polymer has the maximum degradation temperature (370 °C) and it is expected to shift to a lower temperature for natural fibre-reinforced composites. This is because the insertion of lower thermal stability natural fibre would cause early thermal degradation, due to the lignin and hemicellulose components in fibre. The presence of JSP in the PLA destabilised the PLA matrix in the composite whereby some portion of the polymer is replaced by less thermally stable fibres in the composite materials [36].

When looking into the effect of bleaching treatments, a higher onset temperature was found compared to untreated fibre composite specimens. This shows the success of bleaching treatment regarding the removal of low-thermal stability non-cellulosic components on the fibre’s surface, improving the fibre’s overall thermal properties. However, as expected, JSP specimens (10JSP and 30JSP) had a higher mass residue at the end of the analysis, due to the char formation by the lignin constituent. It provides better dimensional integrity for untreated fibre composites. Figure 9 shows TGA and Derivative thermogravimetric analysis (DTGA) curves for thymol insertions in PLA and 30BSJP composites. It found insignificant changes for thymol inclusion in the composite since the changes were less than 2.5% for all specimens in this study.

### 3.6. Decomposition in Compost

Different loadings of JSP and BJSP of PLA composites were buried under a soil surface that consisted of compost and garden soil. The soil environment contains different kinds of controlled microorganism which help in the composite’s degradation process. Figure 10 shows the observation for PLA and its composites containing 10 and 30 wt% of JSP or BJSP reinforcement that were removed from controlled composting conditions for day 7 and 15. From the table, the degradation of PLA is very slow in soil by which there are no significant changes of colour or mass losses. This may be due to the contaminated sample which interrupts the microorganism to attack and metabolize the polymer matrix. It can be shown here that PLA degradation upon disposal in the environment (environmental degradation) is more challenging because PLA is largely resistant to attack by microorganisms in soil or sewage under ambient conditions [37].

The observation on PLA composites was expected as the higher amount of fibre content increased the rate of biodegradation. Microorganisms were attracted to the fibre in the composite. They consumed the fibre and caused a fracture in the PLA chains. However, the 30BJSP composite shows some of the changes in colour with an insignificant reduction in mass losses. This may be due to the stronger interfacial adhesion synchronised with mechanical properties, which lower the number of voids that are exposed to enzyme hydrolysis and hence resulted in a longer degradation time [38].

### 3.7. Antimicrobial Activity

Figure 11 shows the zone of inhibition of active PLA, 30JSP and 30BJSP composites with 15 wt% thymol. The antimicrobial activity was tested in vitro to determine the potential of thymol as an antimicrobial agent in the composites. The 30BJSP–15THY composite had a zone with *S. aureus* present but it was small while 30JSP–15THY showed a clear zone of inhibition against *S. aureus*. The findings in this study show antimicrobial activities from thymol insertion, similar with a previous study [23] where 10 wt% thymol and 30 wt% kenaf fibre displayed a clear zone of inhibition against *E. coli*. In this study, a brittle and less-rigid composite such as 30JSP-15THY was able to release the thymol significantly and inhibit the growth of Gram-positive bacteria than the bleached-fibre composite.

## 4. Conclusions

The objective in this study has been achieved—i.e., to fabricate a low-cost natural fibre-reinforced polymer that can be potentially applied to packaging. The insertion of jackfruits fibres has replaced a portion of expensive PLA raw materials, in turn reducing the overall cost and 30 wt% of fibre insertion recorded the highest tensile performances. Besides, the bleaching treatment helped to improve the composite’s performance in terms of its mechanical and thermal properties, as well as enable it to last for a longer service period. Moreover, 30JSP–15THY was able to release the thymol significantly and inhibit the growth of Gram-positive bacteria than the bleached-fibre composite. As a further development, 30 wt% of the bleached-fibre insertion composite should receive further analysis since it has high potential to reduce the cost of bioplastic products with a minimum alteration of overall performances.

## Figures and Tables

**Figure 1 polymers-12-02622-f001:**
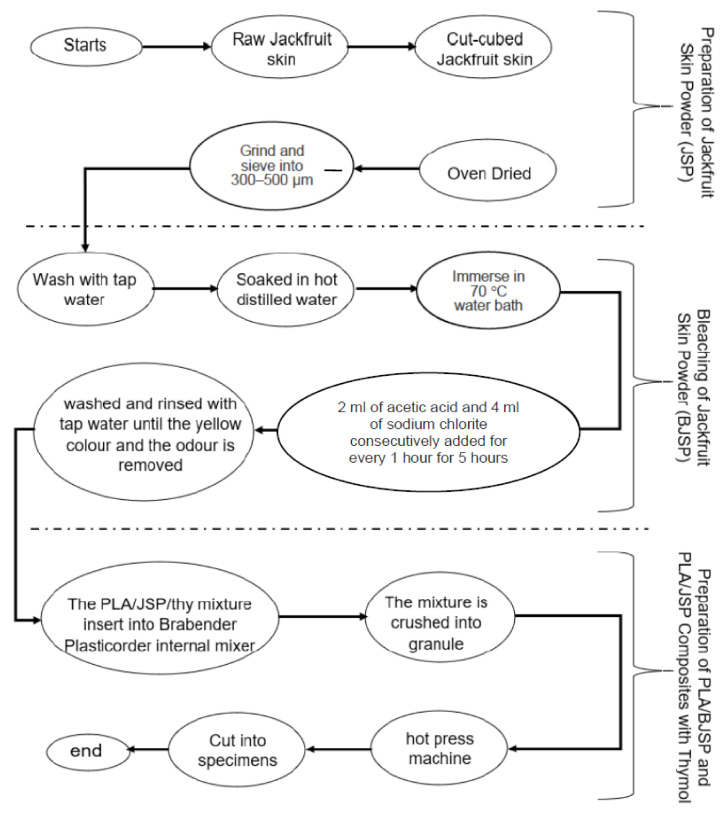
Flow diagram of the current study’s methodology.

**Figure 2 polymers-12-02622-f002:**
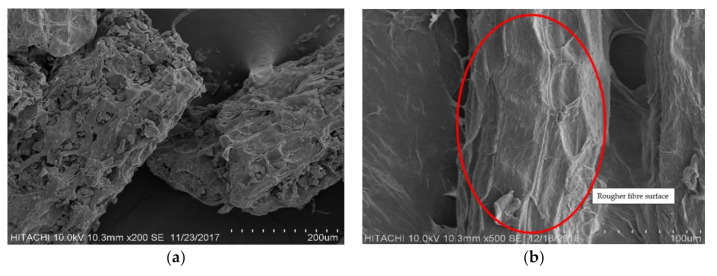
Scanning electron microscopy (SEM) micrograph for (**a**) unbleached jackfruit skin powder (JSP) and (**b**) bleached jackfruit skin powder (BJSP).

**Figure 3 polymers-12-02622-f003:**
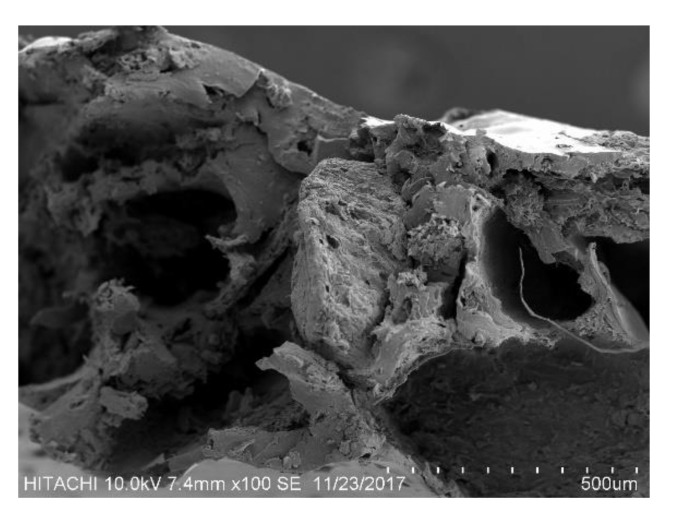
SEM micrograph for 10 wt% thymol in 30BJSP composite.

**Figure 4 polymers-12-02622-f004:**
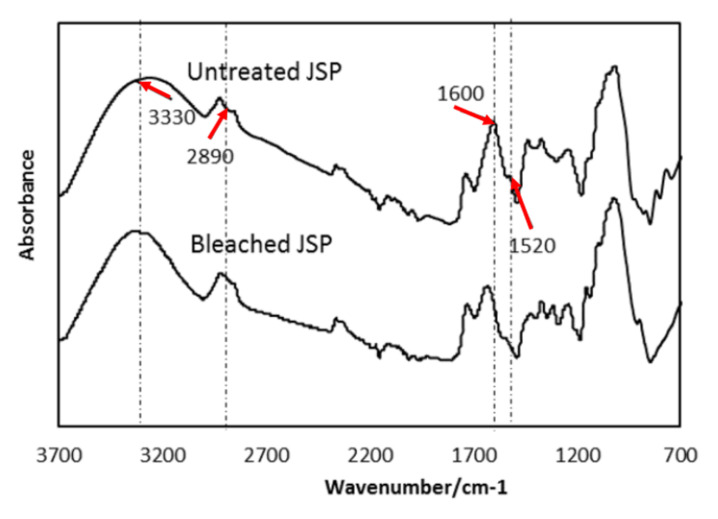
Fourier-transform infrared (FTIR) spectrum for JSP and BJSP.

**Figure 5 polymers-12-02622-f005:**
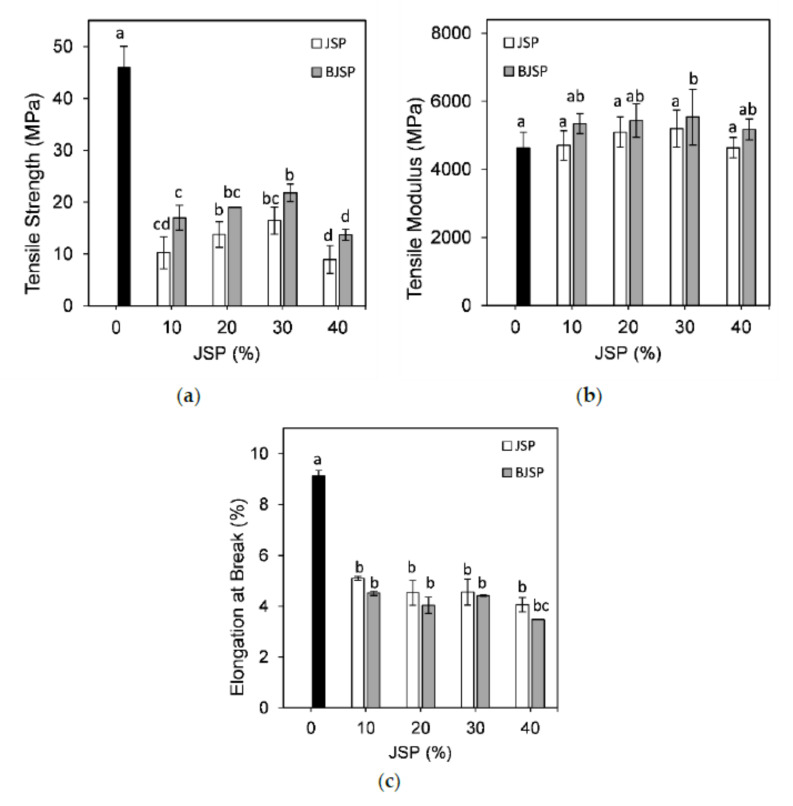
The figure shows the (**a**) tensile strength, (**b**) tensile modulus and (**c**) elongation at break of PLA and its composites. Data represents mean ± standard deviation of four independent repeats. Different letters in each JSP loading indicate significant differences (*p* < 0.05).

**Figure 6 polymers-12-02622-f006:**
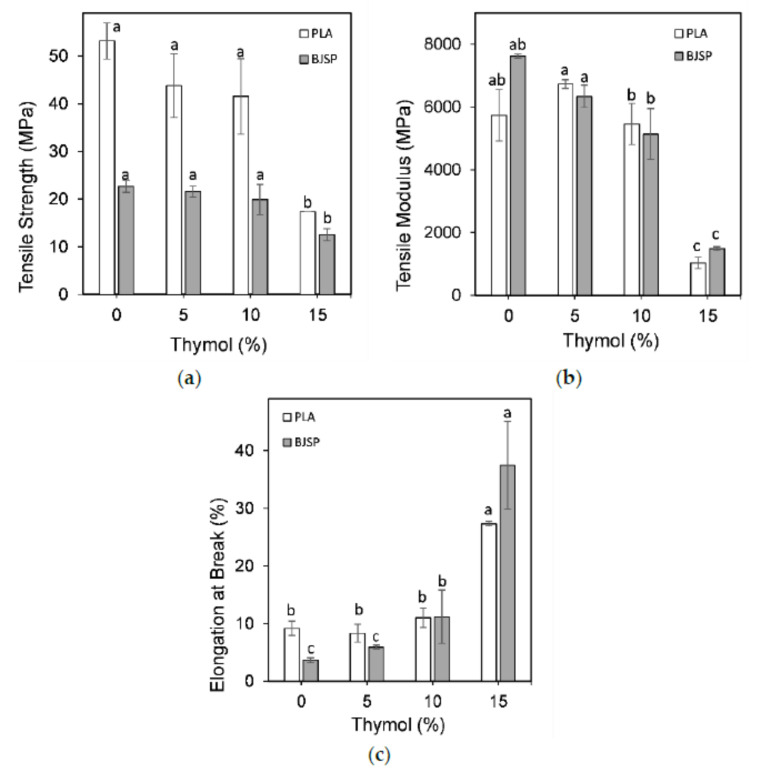
The figure shows the (**a**) tensile strength, (**b**) tensile modulus and (**c**) elongation at break of PLA- and 30BJSP-based composites with different thymol contents. Data represents mean ± standard deviation of four independent repeats. Different letters in each thymol loading indicate significant differences (*p* < 0.05).

**Figure 7 polymers-12-02622-f007:**
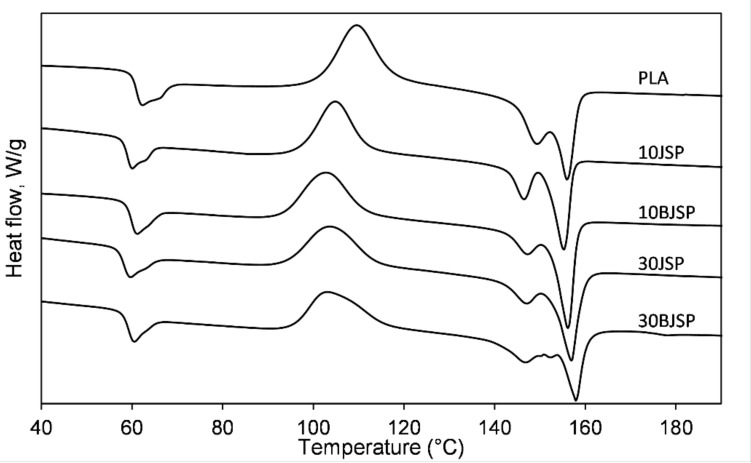
Differential scanning calorimetry (DSC) analysis for PLA and its thymol composites.

**Figure 8 polymers-12-02622-f008:**
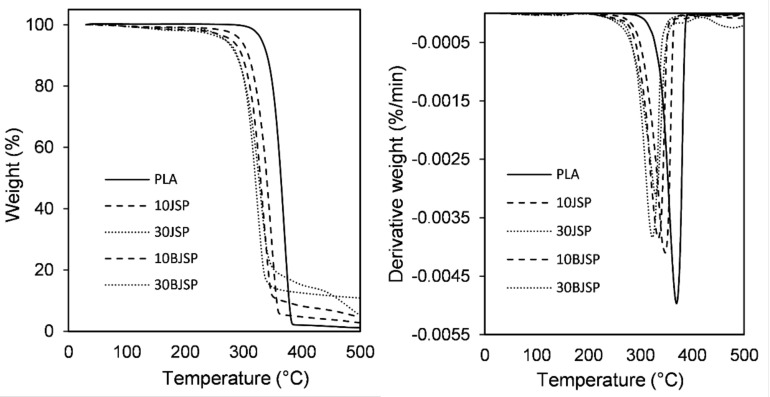
PLA and its composites’ curve for (**a**) thermogravimetric analysis (TGA), (**b**) derivative thermogravimetric analysis (DTGA).

**Figure 9 polymers-12-02622-f009:**
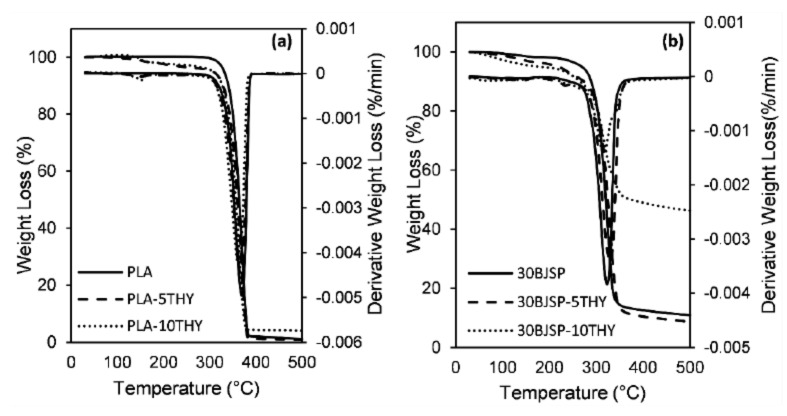
TGA and DTGA curve, (**a**) PLA and its thymol composites and (**b**) 30BSP and its thymol composites.

**Figure 10 polymers-12-02622-f010:**
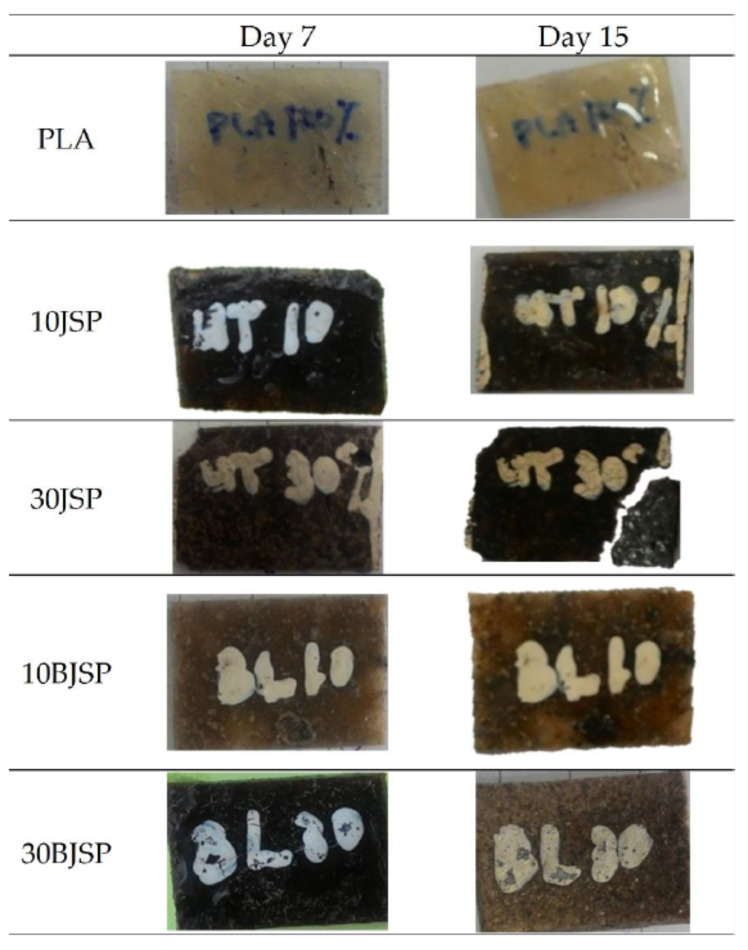
Soil biodecomposed appearance for PLA and its composite.

**Figure 11 polymers-12-02622-f011:**
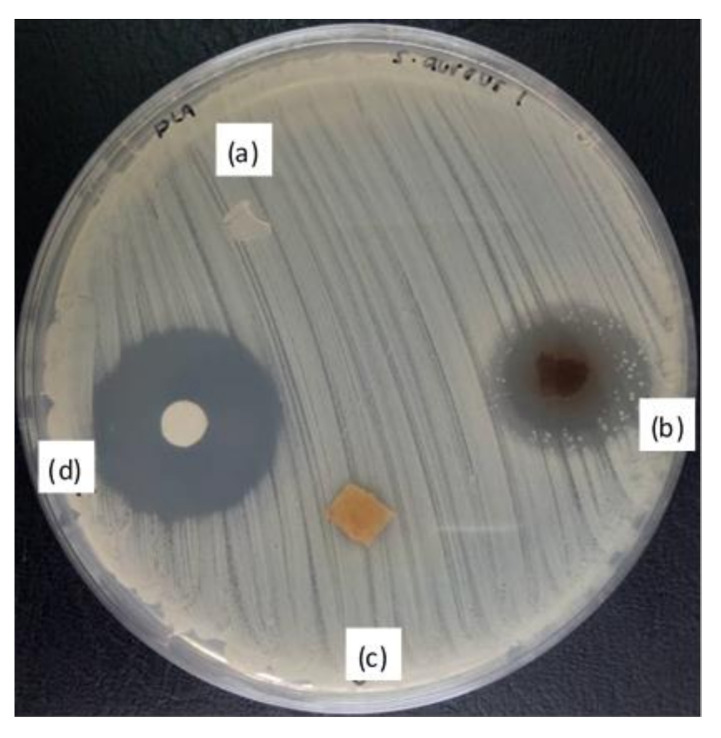
Antimicrobial activity of PLA-based composites against *S. aureus* after 24 h of incubation at 37 °C for (**a**) PLA-15THY, (**b**) 30JSP-15THY, (**c**) 30BJSP-15THY and (**d**) negative control.

**Table 1 polymers-12-02622-t001:** Formulation of poly(lactic acid) (PLA) and PLA composites.

Composite	Fibre Powder Content (wt%)	Naming	Composite	Fibre Powder Content (wt%)	Thymol Content (wt%)	Naming
PLA	0	PLA				
Untreated JSP	10	10JSP	PLA-THY	0	5	PLA-5THY
	20	20JSP		0	10	PLA-10THY
	30	30JSP		0	15	PLA-15THY
	40	40JSP				
Bleached JSP	10	10BJSP	30BJSP-THY	30	5	30BJSP-5THY
	20	20BJSP		30	10	30BJSP-10THY
	30	30BJSP		30	15	30BJSP-15THY
	40	40BJSP				

**Table 2 polymers-12-02622-t002:** Thermal analysis parameters obtained from ifferential scanning calorimetry (DSC) thermograms of PLA and its composites.

Sample	*T*_g_ (°C)	*T*_cc_ (°C)	*T*_m1_ (°C)	*T*_m2_ (°C)	Δ*H*_cc_ (J/g)	Δ*H*_m_ (J/g)
PLA	59.69	101.99	151.27	155.58	21.72	32.63
10JSP	57.69	98.17	150.74	154.99	15.16	27.85
10BJSP	58.41	95.01	151.12	155.55	14.00	36.46
30JSP	56.66	95.39	151.24	156.19	15.45	32.71
30BJSP	57.61	96.76	152.53	157.13	13.41	33.34
PLA-5THY	51.60	93.87	145.72	151.50	14.94	25.96
PLA-10THY	40.38	86.96	140.05	147.27	16.47	23.42
30BJSP-5THY	50.10	-	145.83	-	-	42.28
30BJSP-10THY	39.50	-	129.79	-	-	36.18

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
