# Peer review of "The Effect of Jackfruit Skin Powder and Fiber Bleaching Treatment in PLA Composites with Incorporation of Thymol"

_polymers, 2020, doi:10.3390/polym12112622_

Round 1

Reviewer 1 Report

please see my comments in the attached file

Author Response

Response to reviewer’s comments

  1. Overall, the introduction is well presented. The state of the art is given and the aim of the study is clearly defined.

We would like to express our gratitude to the reviewer for taking your valuable time to review our manuscript. 

  1. please mention its botanical name (latin name)

The botanical name of Jackfruits, Artocaprus heterophyllus, has been highlighted in the manuscript.

  1. please make clearer why you used thymol and its main characteristics. I would appreciate if you can offer some alternatives (instead of using thymol)

More explanation on the reason of selecting thymol and some other alternative antimicrobial agents were suggested in manuscript.

  1. add reference at line 90

References have been added into the manuscript to show the gap between current study and previous studies.

  1. This is an excellent session. Please add a flow diagram to show us (readers) the experimental procedure

We have inserted the flow diagrams in the methodology section.

  1. Overall, it is an excellent discussed session. Please pay attention to my comment for subsession 3.3.

We would like to express our appreciation to reviewer to handling our manuscript.

  1. i suggest to use colour!!!

The circle has been changed into red colour and bolded.

  1. I suggest coloured arrows to be used. also omit the (a) from the right

The FTIR figure has changed to red colour arrows and also omitted the (a) at the right side.

  1. A well discussed session. However, I would like to see in the data presented in Figures 3 and 4, if there are significant differences between treatments. So, perform a statistical analysis (one-way ANOVA or test)

We have performed the statistical analysis (one-way ANOVA) and inserted in manuscript with some discussions.

Reviewer 2 Report

The article deals with the development of a composite formulation for food packaging based on poly(lactic acid) (PLA). The main objective in this study is to fabricate a low-cost natural fibre reinforced polymer, however, the initial costs to produce bioplastics are typically high, for this reason a tropical fruit waste, such as jackfruit skin, can be used as a cost-reducing filler for PLA.

The paper is a systematic, well-rounded thermal, mechanical and morphological characterization of PLA based biocomposites and, in my opinion, the article is noteworthy of publication but it needs major revisions listed below:

  1. In the Introduction it would be interesting, as far as antimicrobial treatments on polymers used for packaging are concerned, to outline other additives (such as chitin nanofibrils) in addition to the tested thymol and to make a small comparison.

  1. In figure 3a when the mechanical properties of pure PLA are shown it makes no sense to show a double column in the histogram with the same values, as there is no presence of the two different fibres

  1. In the literature, very often, as the content of natural fibres increases, which are randomly dispersed within the polymer matrix, analytical models are shown alongside the experimental data that tend to predict the mechanical behaviour of the biocomposites themselves. It would be interesting, for example for elastic modulus or stress at break, to understand what kind of behaviour biocomposites follow in this paper. With regard to the elastic modulus, do the biocomposites in this paper follow the rule of mixture? Is the effect of the filler exponential? Please explain better.

  1. Table 1: It will be interesting to evaluate the crystallinity percentage of the pure polymer and the composites and to give a better explanation of mechanical properties through the relationship with the potential change in crystallinity.

  1. To corroborate and make more reliable the discourse on the adhesion and dispersing agent effect of thymol it is necessary to show SEM images of the fractured sections of biocomposites.

PLA 7001D is typically used for blow molding, why the choice of this starting polymer? Is jackfruit a natural fibre that can attend this purpose as a dispersed phase in PLA?

Author Response

Reviewer 2's comments 

Thank you for the valuable time to review our manuscript, here are the replies of the comments. 

  1. In the Introduction it would be interesting, as far as antimicrobial treatments on polymers used for packaging are concerned, to outline other additives (such as chitin nanofibrils) in addition to the tested thymol and to make a small comparison.

More explanations on the reason for selecting thymol and some other alternative antimicrobial agents were suggested in the manuscript.

  1. In figure 3a when the mechanical properties of pure PLA are shown it makes no sense to show a double column in the histogram with the same values, as there is no presence of the two different fibres

The figure has revised. The duplicate double column has been removed represented with single column.

  1. In the literature, very often, as the content of natural fibres increases, which are randomly dispersed within the polymer matrix, analytical models are shown alongside the experimental data that tend to predict the mechanical behaviour of the biocomposites themselves. It would be interesting, for example for elastic modulus or stress at break, to understand what kind of behaviour biocomposites follow in this paper. With regard to the elastic modulus, do the biocomposites in this paper follow the rule of mixture? Is the effect of the filler exponential? Please explain better.

We thank the reviewer for this interesting suggestion. However, an analytical model to predict the mechanical behavior of the biocomposites is out of the scope for this current paper.   This would be an interesting topic to be considered in the future in a separate paper.  We note that the reviewer has made this as a suggestion and not a mandatory change to the manuscript. 

  1. Table 1: It will be interesting to evaluate the crystallinity percentage of the pure polymer and the composites and to give a better explanation of mechanical properties through the relationship with the potential change in crystallinity.

We appreciate the suggestion by reviewer to make our manuscript more informative. However, this study’s project has be done and we are unable to carry out anymore analysis.

Yet we agreed that crystallinity is one of the main factors on improving crystallinity. Therefore, we have included some discussions and citations relating composite’s crystallinity index and mechanical properties. We are hoping all readers could acknowledge the importance of the crystallinity index on mechanical properties. Nevertheless, we shall include the crystallinity test in our future studies.

  1. To corroborate and make more reliable the discourse on the adhesion and dispersing agent effect of thymol it is necessary to show SEM images of the fractured sections of biocomposites.

The SEM image for 10wt% thymol in 30BJSP composite has been added into the manuscript with  discussion on the effect of thymol in composite.

  1. PLA 7001D is typically used for blow molding, why the choice of this starting polymer? Is jackfruit a natural fibre that can attend this purpose as a dispersed phase in PLA?

The purpose of the paper is to reduce the cost of PLA particularly for food packaging. Besides, the previous study as stated below, has also used PLA 7001D as matrix for kenaf/thymol composite in food packaging purpose. It was found that the mechanical and thermal properties of PLA grade 7001D was similar to PLA grade 2003D (injection moulding) for food packaging and food serviceware.

 Tawakkal, Intan & Cran, Marlene & Bigger, Stephen. (2016). The influence of chemically treated natural fibers in poly(lactic acid) composites containing thymol. Polymer Composites. 10.1002/pc.24062.

Round 2

Reviewer 2 Report

The manuscript has been implemented with the suggestions provided by the reviewer and I very much appreciate the changes in the text with regard to morphological and mechanical considerations.

In my opinion the text is now ready for publication.